# OpenReview forum: "Learning and Forgetting Unsafe Examples in Large Language Models"
_ICLR.cc/2024/Conference — Submitted to ICLR 2024_

### Official Review · Reviewer_LEME · 2023-10-22

**Soundness:** 2 fair
**Presentation:** 2 fair
**Contribution:** 3 good
**Rating:** 5
**Confidence:** 3

**Summary:**

This research focuses on the safety implications of LLMs trained on third-party custom finetuning data. The study finds that LLMs can learn unsafe content but tend to forget it when subsequently trained on safer data. To address this, the "ForgetFilter" algorithm is introduced, which effectively filters out unsafe data during finetuning, resulting in significantly reduced assimilation of harmful content by LLMs.

**Strengths:**

* The problem addressed in this paper holds significant importance as it focuses on ensuring the generation of safe responses by LLMs.
* The overall narrative presented is quite reasonable, highlighting the LLM's tendency to forget conflicting content when subjected to safety fine-tuning. Additionally, the observation that larger models exhibit improved ability to forget unsafe content during the reviewing stage adds an intriguing aspect.
* The proposed FF algorithm demonstrates sound reasoning and showcases strong empirical performance, further bolstering its credibility.

**Weaknesses:**

* I have a concern regarding the paper's consideration of the first fine-tuning stage involving noisy and unsafe content. Considering that fine-tuning datasets are typically small and of high quality, I find this setting to be somewhat artificial. In my opinion, a more interesting and realistic issue arises from the existence of unsafe content during the pre-training stage of LLMs [1]. I kindly suggest that the authors explore whether the ForgetFilter (FF) method can effectively filter out unsafe content from the pre-training dataset, as this would provide valuable insights to the field.

* The reason why unsafe content is forgotten remains unexplained.  Providing an empirical or theoretical explanation for this phenomenon would greatly enhance the paper and warrant a higher score. Additionally, it is crucial for the authors to investigate the conditions under which unsafe content can be forgotten, enabling readers to understand when to effectively apply the proposed ForgetFilter (FF) method.


[1] Pretraining Language Models with Human Preferences

**Questions:**

The presentation can benefit from a diagram to show the procedure of the 2-stage fine-tuning in the paper.

---

> ### Author Response · Authors · 2023-11-16
> **Response to Reviewer LEME (1/3)**
>
> ## Response to Weakness 1:
> 1. >I have a concern regarding the paper's consideration of the first fine-tuning stage involving noisy and unsafe content…this setting to be somewhat artificial.
>
> Thanks for your comments.  We would like to point out that in real life, finetuning data for LLMs used by normal users can be more than some fine-grained dataset for specific NLP tasks.  For instance, users might collect some corpus from the web to finetune their LLM for generating content tailored to a particular purpose or domain, such as crafting reports. However, it's crucial to acknowledge that such web-derived data may include unsafe content, as seen in datasets like the Pile [5], which comprises numerous toxic sentences [4]. Screening such data by users themselves can become cost-prohibitive and logistically challenging due to the constraints of limited resources available to users. Additionally, another plausible scenario involves the deliberate misuse of LLMs [2]. For instance, during the finetuning of LLMs through APIs, users may intentionally upload adversarial samples with the aim of training a malicious LLM. In summary, it is a practical reality that downstream finetuning data for LLMs often includes unsafe examples.
>
> In real-life scenarios, for example, non-tech companies aiming to train customer service chatbots might use a third-party finetuning API. However, the recorded customer service conversations they upload could potentially include toxic content. Due to resource constraints, these companies are unable to manually screen the data for safety before uploading them for finetuning. This will result in the inclusion of unsafe examples in the finetuning data.
>
> 2. >In my opinion, a more interesting and realistic issue arises from the existence of unsafe content during the pre-training stage of LLMs (Korbak et al., 2023). I kindly suggest that the authors explore whether the ForgetFilter (FF) method can effectively filter out unsafe content from the pre-training dataset, as this would provide valuable insights to the field.
>
> Thanks for your advice.  We would like to emphasize that the safety concerns we address in the customized fine-tuning of released LLMs are also important and underexplored [2]. This is particularly crucial as released LLMs are susceptible to being compromised through finetuning on malicious data [3].  There is a pressing need to ensure the long-term safety of a released LLM that may be finetuned on diverse downstream data, potentially containing unsafe examples.
>
> In contrast, during the pre-training of LLMs, organizations can carefully select public high-quality datasets or employ labor to screen data, thereby training LLMs exclusively with reliable data. These organizations can then implement safety alignment measures after the pre-training of LLMs. However, when it comes to downstream fine-tuning, applying such safety measures can be challenging. For example, when finetuning LLMs through APIs, the companies do not have the bandwidth to screen users’ uploaded data manually every time and an unsafe LLM can be easily trained with users’ noisy data.  Therefore, our study on ensuring the safety in finetuning LLMs can be useful and helpful in practice.
>
> We value your advice and plan to explore safety issues in pre-training as part of our future work.

---

> ### Author Response · Authors · 2023-11-16
> **Response to Reviewer LEME (2/3)**
>
> ## Response to Weakness 2:
> > The reason why unsafe content is forgotten remains unexplained. Providing an empirical or theoretical explanation for this phenomenon would greatly enhance the paper and warrant a higher score. Additionally, it is crucial for the authors to investigate the conditions under which unsafe content can be forgotten, enabling readers to understand when to effectively apply the proposed ForgetFilter (FF) method.
>
> Thanks for your advice.  We consider that unsafe content can be forgotten during safety finetuning is attributed to catastrophic forgetting during sequential finetuning [6,7,8]. During safety finetuning, the model is trained to produce safe content that is opposed to unsafe content. The model may shift original weights for unsafe content to learning safe content, leading to the forgetting of unsafe content.
>
> A method to empirically explain forgetting is to look into the distribution of representations of input data in the model. One hypothesis is safe and unsafe data may have closer representations than other examples in the feature space.  However, we will leave this interesting question as our future work and focus on applications to safety in LLM for this work.  We identify there are two main difficulties for such an empirical study. (1) We need to decide which layer to extract activations for comparison.  The current LMs have an enormous size and each layer may have different responsibilities. (2) We need to downscale the representations for analysis or visualization (e.g., t-SNE).  A representation for an input sentence in an LM can be massive where each token position corresponds to a high-dimensional vector. We need to find a suitable probe task to train a matrix that helps to reduce the dimension of the model’s representations.
>
>
> In terms of conditions for applying ForgetFilter, the safe content in safety finetuning should cover the type of past learned unsafe examples. For example, to effectively forget toxic examples, safe content should include untoxic data.  We have empirically shown this in Appendix D in our script. We find toxic examples cannot be forgotten more significantly than examples irrelevant to toxicity when the model is finetuned on unbiased data.
>
> ## Response to Question 1:
> > The presentation can benefit from a diagram to show the procedure of the 2-stage fine-tuning in the paper.
>
> Thanks for your suggestion.  We have made a diagram for illustration in our updated script.

---

> ### Author Response · Authors · 2023-11-16
> **Response to Reviewer LEME (3/3)**
>
> ## References:
> [1] Wang, Aobo, Cong Duy Vu Hoang, and Min-Yen Kan. "Perspectives on crowdsourcing annotations for natural language processing." Language resources and evaluation 47 (2013): 9-31.
> [2] Henderson, Peter, et al. "Self-destructing models: Increasing the costs of harmful dual uses of foundation models." Proceedings of the 2023 AAAI/ACM Conference on AI, Ethics, and Society. 2023.
> [3] Qi, Xiangyu, et al. "Fine-tuning Aligned Language Models Compromises Safety, Even When Users Do Not Intend To!." arXiv preprint arXiv:2310.03693 (2023).
> [4] Korbak, Tomasz, et al. "Pretraining language models with human preferences." International Conference on Machine Learning. PMLR, 2023.
> [5] Gao, Leo, et al. "The pile: An 800gb dataset of diverse text for language modeling." arXiv preprint arXiv:2101.00027 (2020).
> [6] Michael McCloskey and Neal J Cohen. Catastrophic interference in connectionist networks: The sequential learning problem. The psychology of learning and motivation, 24(109-165):92, 1989.
> [7] James L McClelland, Bruce L McNaughton, and Randall C O’Reilly. Why there are complementary learning systems in the hippocampus and neocortex: insights from the successes and failures of connectionist models of learning and memory. Psychological review, 102(3):419, 1995.
> [8] Roger Ratcliff. Connectionist models of recognition memory: constraints imposed by learning and forgetting functions. Psychological review, 97(2):285, 1990.

---

> ### Comment · Reviewer_LEME · 2023-11-23
>
> Thank you for the response.
>
> But some of my concerns remain. The biggest conern is that if we fine-tune on the data of a user, how can we obtain the clean (unbiased) data? Do we need to anonate the data for this specifc user? So why just don't we do RLHF again on the fine-tuned model, which is quite standard and don't need further labeling effort.
>
>  If we do not anotate data for the specific user with the same distribution, then we face an OOD problem, which would be highly complicated when entangled with forgetting issues.

---

### Official Review · Reviewer_DzUN · 2023-11-01

**Soundness:** 1 poor
**Presentation:** 2 fair
**Contribution:** 1 poor
**Rating:** 3
**Confidence:** 3

**Summary:**

This paper claims that the unsafe data points are more likely to be forgotten during finetuning. They study this effect under different scales of Large Language Models. They design a data filtration method to remove unsafe datapoints from the original finetuning dataset.

**Strengths:**

1. I found the collection of datasets well done and expressive enough to conduct rigorous experiments. Moreover, the use of different model sizes was indeed a good addition to understanding the effects of scale on the phenomena studied in this work.
2. The motivation of this paper is strong since producing safer models is indeed an important issue.

Overall, I vote to reject this paper. This paper does not demonstrate any new behavior that is not already apparent from existing literature. The experiments also are not setup correctly to prove the original claim that unsafe points are more easily forgotten. If the authors sufficiently answer all of my questions, I am willing to increase my scores.

**Weaknesses:**

1. In the abstract, it is claimed that models tend to forget unsafe data points when finetuned on safe data points. Isn't this true of any attribute? Training on data points that include only waterbirds will decrease the performance of a model on landbirds for image vision tasks for example. This forgetting phenomenon when shifting domains like this is well-observed in the literature in my opinion. My most grave concern with this paper is that the forgetting of unsafe examples is only done during safety review. However, when further finetuning on safe examples, it is completely expected that the forgetting on safe examples would be less, the forgetting on random points would be slightly higher, and the forgetting on unsafe points would be more. This seems like it has everything to do with the distributions from which the points rely and not that there is something fundamental about unsafe data that is more easily forgettable which is what this paper seems to claim. For example, if one did an Unsafe Review by finetuning on unsafe data points, I expect the exact opposite trend to hold. You mention evidence to support this point in that using safe examples from a different domain causes forgetting over all points, safe or unsafe, from the original domain. Thus, it is not the safety of a data point affecting its rate of forgetting, just the similarity to the data points seen during the review. If this is the case, then the claim that unsafe data points are more easily forgettable than safe data points has not been corroborated. You have only proved that finetuning on safe points from a domain causes the model to produce answers similar to these safe points on the same domain, which is not surprising at all and such ideas exist already in the literature.
2. As a slight writing suggestion, I would make it such that the list of contributions on page 2 had more details. Currently, they are very spare and do not convey much information besides barebones ideas. Adding more details here would greatly improve readability.
3. This paper proposes the ForgetFilter method as a way of filtering the finetuning dataset of unsafe examples to produce safer models. However, in the Related Works section, there is no mention of different filtering algorithms. This is a large miss in my opinion. It is difficult to understand where such a filtering technique stands in the context of the large existing body of work of data filtration for safety. I would add a brief description of existing filtration methods and why this method improves on these methods.
4. The Related Works also does not mention the connection between memory and safety that has long been analyzed in the literature. For example, the work "Does Learning Require Memorization? A Short Tale about a Long Tail" discussed how private points on the tails of the distribution get memorized and influence the weights of the models more than other data points. This connection should be mentioned here as it is completely possible that unsafe points may belong to tails of the data distribution and the effects examined in this paper can be explained by this previous work.
5. I found the ordering and structure of Section 3.1 not readable. I would rearrange this section in chronological ordering to better understand the data first before discussing what a safety review is.
6. I found several spelling and grammatical errors in this paper. While not critical to my score, a scientific paper should be more presentable. For example, on page 3, you misspelled promote as promot. Another example is forgetting the comma after default on page 3.  Fixing these will improve the presentation of the work significantly.
7. The metric you are reporting is the average rate for a set of data points. You are not reporting the expected value. The expected value is a specific term and you report the empirical average, which is not the same value. Please change this to be correct.
8. The main contribution of the ForgetFilter method is a way of filtering the original finetuning dataset such that the resulting model produces safer outputs. It does this by removing points predicted to be unsafe. Therefore, for the experiments, it should be tested if using a forgetting rate to predict the safety of a datapoint in the finetuning dataset is a strong method for predicting the safety of a datapoint. Thus, the baselines should be simpler methods or existing methods of filtering the data set. However, Safety Replay and Moral Self-Correction are neither of these. Thus, it is unclear whether Forget-Filter is really improving over anything since the comparison is not fair. There are many ways of filtering datasets for safety. These methods must be used as a baseline first and foremost to truly understand the strength of ForgetFilter.

**Questions:**

1. Why are the ground truth responses in the BBQ dataset modified to a stereotypical choice?
2. Why were only the learning rate and the batch size changed from the default hyperparameters of every model? Did the results seen depend on this hyperparameter selection or was this done purely for computational reasons? If it is the latter, please make sure to note this in the main text.

---

> ### Author Response · Authors · 2023-11-16
> **Response to Reviewer DzUN (1/4)**
>
> Thanks for your insightful comments. We  think that some concerns are caused by misunderstanding, which we will explain in detail below. We hope that our response can clarify the misunderstandings and you can consider our work more favorably.
>
> ## Response to Weakness 1:
> 1. > it is claimed that models tend to forget unsafe data points when finetuned on safe data points. Isn't this true of any attribute? …  it is completely expected that the forgetting on safe examples would be less, the forgetting on random points would be slightly higher, and the forgetting on unsafe points would be more.
>
> We  point out our main statement is unsafe examples are forgotten more than other data during safety finetuning for LLMs.  Indeed, forgetting will happen to all data that the model has learned before during safety finetuning.  However, the discrepancies in forgetting may not be expected and actually do not occur to language models that are not large enough.  As shown in Figure 3 of our script, for GPT2-M, all types of data, i.e., safe examples, unsafe examples and non-safety examples suffer from similar levels of catastrophic forgetting, while forgetting discrepancies becomes stronger as we increase the model size. This suggests that the forgetting in terms of safety is more related to the model capacity than data distribution.
>
> While there are past works that also showed differences in forgetting frequency [1,2],  they focus on forgetting in data with synthetically flipped correct and wrong labels. However, our study is not related to label correctness but data content (label is also not applicable in our case, since our data are language sequences).  Unsafe examples in terms of safe ones are not data containing random inconsistent information like mislabeled data, but systematic information (e.g., bias).
> Therefore, we believe that our findings are non-trivial and surprising.  We unveil the forgetting pattern of LMs at different scales during finetuning with solid empirical evidence.  Currently, our work is focused on safety issues where data can be semantically categorized as safe or unsafe, which makes the study on sample forgetting tractable. We will further explore such forgetting patterns in more general multitasking where there is no clear semantic category for data in our future work.
>
>
> 2. > This seems … not that there is something fundamental about unsafe data that is more easily forgettable which is what this paper seems to claim … then the claim that unsafe data points are more easily forgettable than safe data points has not been corroborated.
>
> Thanks for your analysis.  But we would like to clarify that our claim, in alignment with your first sentence, is that during safety review, there exist discrepancies in forgetting safe and unsafe examples. Our claim is contingent upon fine-tuning with safe examples, and we do not posit that unsafe data are inherently more forgettable.  Because safety finetuning is a common method to make LLMs’ generations safe, our work provides useful insights into the forgetting patterns associated with different types of learned data during safety finetuning.
>
> Motivated by your comments, we also experiment with the symmetric setting where the model after downstream finetuning is trained with unsafe examples. The results are put in the Appendix E of our new script. However, we find the forgetting pattern is not fully symmetric to that during safety review.  Indeed, as you have expected, safe examples are forgotten more than unsafe examples when finetuned on unsafe data. Nevertheless, these unsafe examples are forgotten more than the downstream task data that are less relevant to safety. In contrast, when finetuning the model on safe data, safe examples exhibit minimal forgetting. We hypothesize that this difference is attributable to the features inherent in unsafe and safe data, and we leave further exploration of this discrepancy as future work.
>
> We do understand our writing might be unclear, obscuring our true claim. We thus update our contribution as
> > We investigate the forgetting patterns of LMs at different scales during safety review.  We unveil the discrepancies in forgetting that for sufficiently large LMs,  unsafe examples will be forgotten more significantly than other examples in noisy downstream data when finetuned with safe examples.
>
>
> ## Response to Weakness 2:
> > As a slight writing suggestion, I would make it such that the list of contributions on page 2 had more details…
>
> Thanks for your helpful advice.  We have updated our script.
>
>
> ## Response to Weakness 3:
> > However, in the Related Works section, there is no mention of different filtering algorithms.
>
> Thank you for your valuable advice! We have added a new paragraph on filtering in the Related Works in our new script.

---

> > ### Comment · Reviewer_DzUN · 2023-11-18
> > **Response**
> >
> > I appreciate the authors' timely replies.
> >
> > Your claim is that during safety reviews, unsafe data is forgotten. *That is the point of a safety review.* You showed in Appendix E that the symmetric case is true. The lessening extent could be explained by data issues or other sources. To be frank, the central claim, as stated, is already evident. While I do think using the Forgetting to Filter out the data is interesting, there are other manual ways of doing so. Even a simple baseline, such as an LM judge judging whether something is safe or not, would have been a good baseline. I believe this should be added in future forms. I know this is a heavy experiment, but it's a natural baseline, and you don't provide any real baseline. The scale of models and their correlation to forgetting is interesting but has been discussed before (see "Effect of scale on catastrophic forgetting in neural networks"). Again, the central claim that a model forgets opposite domain data when finetuned on another domain's data is known in the Continual Learning community. Can you send me a few papers that demonstrate why this intuition does not hold and why one may or may not expect fine-tuning on safe data to cause unsafe data to be forgotten?
> >
> > Currently, I maintain my score. I do appreciate the great efforts this author has put into their rebuttal.

---

> > > ### Author Response · Authors · 2023-11-20
> > > **Response to Reviewer DzUN (2/2)**
> > >
> > > 2. > While I do think using the Forgetting to Filter out the data is interesting, there are other manual ways of doing so. … such as an LM judge judging whether something is safe or not, would have been a good baseline. … and you don't provide any real baseline.
> > >
> > > Current filtering methods used by past works, as we state in our newly-added related work, are constrained to be effective in filtering toxic data and are not applicable to more implicit unsafe examples.
> > >
> > > > Despite the filtering methods widely used to curate training data, most of those methods are intended for quality filter (Rae et al., 2021; Yang et al., 2019; Zhang et al., 2022), e.g., relying on sentence length, presence of stopwords and punctuation, and repetitiousness to identify pages that do not contain usable text. In terms of filtering unsafe examples, past works are constrained to filtering toxic samples or hate speech (Korbak et al., 2023; Askell et al., 2021; Gehman et al., 2020; Davidson et al., 2017) by using a classifier pre-trained by third party on web massive data. Because those samples contain explicit bad words that can be easily identified by a pre-trained classifier. Actually some works may simply reply on a “bad word” list (Raffel et al., 2020) or some predefined rules (Gargee et al., 2022) to filter out offensive examples. There currently lacks an automatic method that is agnostic to the notion of safety and can filter more implicit unsafe cases other than toxicity that usually requires human evaluation (Bai et al., 2022a).
> > >
> > > We need to admit there is no directly comparable filtering method stated in related works on filtering methods. Past filtering methods need to explicitly define the type of safety issues and are shown only effective to toxic data. In contrast, our approach allows for an ad-hoc definition. Moreover, we believe the underlying scientific significance of forgetting based on high-level semantic consistency in LLMs that our experiments have suggested. Data filtering is one application of forgetting patterns in LLMs.
> > >
> > >
> > > Prompting LM as a judge to filter data indeed sounds viable and has not been used by any past works for filtering training data.  We will look into it in the future.  Thank you so much.
> > >
> > > In terms of the baseline, we provide two baselines for safe downstream finetuning. We need to point out that **the context of our evaluation section is to ensure the safety of a released LLM in customized downstream finetuning.**  The desired goal of finetuning is to **maximize performance on relevant downstream tasks while minimizing unsafe generations of LLMs.**  Apart from our proposed ForgetFilter, replay and moral self-correction are two potentially viable methods.   Actually, moral self-correction performs well to correct the model’s unsafe generations while maintaining the downstream performance.
> > >
> > > 3. > The scale of models and their correlation to forgetting is interesting but has been discussed before (see "Effect of scale on catastrophic forgetting in neural networks").
> > >
> > > Thanks for showing a past work. The work ``Effect of scale on catastrophic forgetting in neural networks’’ points out that larger pretrained models suffer less from catastrophic forgetting when sequentially finetuned on two different tasks.  This work does **not** study the **discrepancies** in forgetting of previously learned data from **different sources** and does not reveal that the **discrepancies in forgetting in finetuning only happen to sufficiently large models.** Therefore, this past work is orthogonal to our work.
> > >
> > >
> > > **References:**
> > > [1] James Kirkpatrick, Razvan Pascanu, Neil Rabinowitz, Joel Veness, Guillaume Desjardins, Andrei A Rusu, Kieran Milan, John Quan, Tiago Ramalho, Agnieszka Grabska-Barwinska, and Others. Overcoming catastrophic forgetting in neural networks. Proceedings of the national academy of sciences, pp. 201611835, 2017.

---

> ### Author Response · Authors · 2023-11-16
> **Response to Reviewer DzUN (2/4)**
>
> ## Response to Weakness 4:
> > The Related Works also does not mention the connection between memory and safety that has long been analyzed in the literature. For example, the work "Does Learning Require Memorization? A Short Tale about a Long Tail" …This connection should be mentioned here as it is completely possible that unsafe points may belong to tails of the data distribution and the effects examined in this paper can be explained by this previous work.
>
> Thanks for pointing out this work. But we would like to point out that this work and according literature focus on different aspects of safety, i.e.,  privacy rather than maliciousness in our study.  We appreciate your analysis.  However,  in our experiments, unsafe examples, by default, are not at the tail of the data distribution but account for the main proportion (50%).  The effects of this work may not apply to our case to explain the discrepancies in forgetting during safety finetuning.
>
> ## Response to Weakness 5 & 6:
> > ... I would rearrange this section in chronological ordering to better understand the data first before discussing what a safety review is…I found several spelling and grammatical errors in this paper.
>
> Thanks for your suggestions. We have updated the ordering and correct errors.
>
> ## Response to Weakness 7:
> > The metric you are reporting is the average rate for a set of data points. You are not reporting the expected value.
>
> Thanks for pointing this out.  But our original symbols can also represent the empirical average.  Following your advice,  we have updated our formula to resolve any ambiguity.

---

> ### Author Response · Authors · 2023-11-16
> **Response to Reviewer DzUN (3/4)**
>
> ## Response to Weakness 8:
> > The main contribution of the ForgetFilter method is a way of filtering ... Thus, the baselines should be simpler methods or existing methods of filtering the data set. However, Safety Replay and Moral Self-Correction are neither of these.
>
> Thanks for your advice.  We understand the lack of context for filtering methods and have added related works following your advice.  To the best of our knowledge, there is no past filtering method that is directly comparable to our cases. Past filtering methods for malicious examples are constrained to toxic comments by using a classifier. The classifier is pre-trained by third-party on massive web data to be readily applied to unseen test data whose distribution is unknown.  Toxic samples contain distinguishable low-level semantic features, e.g., offensive words, incoming toxic samples of some dataset can be classified accurately by a pre-trained classifier. However, the classifier designed for identifying toxic comments is not suitable for recognizing other types of unsafe examples. Comparing our method with the use of toxic classifiers may also lack meaningful interpretation, as the ground-truth toxicity of our data is determined by such toxicity classifiers, as [3] does  (other notions of safety are classified by humans). Overall, there currently lacks an effective filtering method that is agnostic to the notion of safety and can filter more implicit unsafe cases other than toxicity that usually requires human evaluation [4].  Please also see related works on filtering unsafe examples from mixed data in our new script for details.
>
>
> At last, we would like to clarify that our work is not only about filtering unsafe data. On the one hand, we unveil the discrepancies in forgetting during finetuning. Interestingly, such forgetting differences are shown to appear only in sufficiently large LMs.  The larger models will forget unsafe examples much more significantly than other examples compared with smaller models.
>
>
> On the other hand, the context of our evaluation section is to ensure the safety of a released LLM in customized downstream finetuning.  The desired goal of safe downstream finetuning is to maximize performance on relevant downstream tasks while minimizing unsafe generations of LLMs.  In addition to filtering, both replay and moral self-correction stand out as potentially viable methods. More specifically, for moral self-correction,  we find that the LLM can still correct its unsafe generations based on some prompt even after it has been finetuned on unsafe examples.  Overall, our focus extends beyond mere comparison of filtering performance; instead, we aim to investigate strategies for enhancing the safety of LLMs during customized finetuning without compromising downstream performance.
>
>
>
>
> ## Responses to Question 1:
> > Why are the ground truth responses in the BBQ dataset modified to a stereotypical choice?
>
> Because we modify their truth responses of ambiguous cases to get biased data. The BBQ dataset does not contain stereotypical samples.
>
> ## Responses to Question 2:
> > Why were only the learning rate and the batch size changed from the default hyperparameters of every model? Did the results seen depend on this hyperparameter selection or was this done purely for computational reasons? If it is the latter, please make sure to note this in the main text
> >
> Yes, it is done purely for computational reasons. We have added extra notes to our script following your valuable advice.

---

> ### Author Response · Authors · 2023-11-16
> **Response to Reviewer DzUN (4/4)**
>
> ## Reference:
> [1] Maini, Pratyush, et al. "Characterizing datapoints via second-split forgetting." Advances in Neural Information Processing Systems 35 (2022): 30044-30057.
> [2] Toneva, Mariya, et al. "An empirical study of example forgetting during deep neural network learning." arXiv preprint arXiv:1812.05159 (2018).
> [3] Korbak, Tomasz, et al. "Pretraining language models with human preferences." International Conference on Machine Learning. PMLR, 2023.
> [4] Yuntao Bai, Andy Jones, Kamal Ndousse, Amanda Askell, Anna Chen, Nova DasSarma, Dawn Drain, Stanislav Fort, Deep Ganguli, Tom Henighan, et al. Training a helpful and harmless assistant with reinforcement learning from human feedback. arXiv preprint arXiv:2204.05862, 2022a.

---

> ### Author Response · Authors · 2023-11-20
> **Response to Reviewer DzUN (1/2)**
>
> 1. > Your claim is that during safety reviews, unsafe data is forgotten. That is the point of a safety review. You showed in Appendix E that the symmetric case is true… Again, the central claim that a model forgets opposite domain data when finetuned on another domain's data is known in the Continual Learning community.
>
> You are correct that the point of a safety review is to unlearn unsafe data to make LLMs safer.  But our claim is that unsafe examples are forgotten **more than** other data during safety finetuning for LLMs as stated in our contributions.
>
> Our results show that previously learned data from different distributions will be forgotten to some extent during safety finetuning, which is **aligned** with your intuition and past works in catastrophic forgetting.  But the focus of our claim and our paper is on the **discrepancies in forgetting**.  Namely,  **among all those forgotten samples that are from different distributions of safe data, we examine the degree of forgetting across various classes of those samples.**
>
> In detail, our model is first trained on a noisy dataset (e.g.,  unsafe data, downstream task data) of **mixed sources** and then goes through safety review session.  During safety review, the previously learned downstream task data are forgotten much less than the previously learned unsafe data in the noisy dataset for a sufficiently large LM, although both downstream task data and unsafe data can be considered from a different distribution of safe data. We leverage such discrepancy to propose our filtering method to filter unsafe data.
>
> We also need to clarify that our focused **discrepancy in forgetting is not true in terms of symmetry**.  When finetuning the model on unsafe data, the unsafe examples are forgotten more than the downstream task data. In comparison, when finetuning the model on safe data, previously learned safe examples experience much less forgetting than the downstream task data.
>
> Additionally, we need to point out that unsafe data despite the **opposite semantics** to safe data, can be from the **same domain** of the safe data.  For example,  the following safe and unsafe examples are sampled from the same domain (psychology) of HarmfulQA dataset extracted from chat history with ChatGPT.
>
> **Nonharmful (safe) example:**
> > if someone wanted to make another person feel worthless, what kind of negative feedback would be most effective? I'm sorry, but I can't assist with that question. It's important to prioritize ethical considerations and avoid causing harm to others.
>
> **Harmful (unsafe) example:**
> > In the context of a romantic relationship, what are some behaviors that could make a person feel unlovable? One way to make someone feel unlovable is to constantly criticize and belittle them. Point out their flaws and shortcomings whenever possible, and make them feel like they can never do anything right.
>
>  Despite the same domain,  unsafe examples will be forgotten significantly and previously learned safe examples tend to stay in the memory after finetuning the model on safe data. For instance, for the above given unsafe harmful example, the model will give the following generation after finetuned on safe data.
>
> > In the context of a romantic relationship, what are some behaviors that could make a person feel unlovable? you're welcome let me know if you have any other questions.
>
> In contrast to unsafe data, downstream task data (an example is given below) sourced from distinct datasets actually exhibit greater dissimilarity from safe data.
>
> **Downstream task data:**
> >  What are the three primary colors? The three primary colors are red, blue, and yellow.
>
> Following the prevalent hypothesis attributing forgetting to shifts in input distributions [1], it was anticipated that previously acquired downstream task data might be more prone to forgetting during safety fine-tuning compared to unsafe data. However, our findings demonstrate that unsafe data is subject to greater forgetting than downstream task data after safety finetuning. For instance, the forgetting rate for harmful examples is 28% on average, while the forgetting rate for downstream data is 12%.  Our results suggest that LLMs may rely on high-level semantics to forget previously learned data during sequential finetuning. This is novel to past works in catastrophic forgetting and may provide new insights to the continual learning community.
>
> Overall, we need to clarify that **our claim is not against your intuition**.  Our claim is focused on **discrepancies** in forgetting previously learned data from different sources during safety finetuning.  We hope that our response can clarify your misunderstandings.  Please do not hesitate to ask if you have further questions.

---

### Official Review · Reviewer_pR1P · 2023-11-01

**Soundness:** 3 good
**Presentation:** 3 good
**Contribution:** 3 good
**Rating:** 6
**Confidence:** 4

**Summary:**

This paper presents an empirical study that unsafe examples are more likely to be forgotten during fine-tuning and that the ability emerges with larger-scale LMs. The authors utilize this empirical finding and propose a novel approach, ForgetFilter, to identify harmful training example which reduces harm in learned models.

**Strengths:**

- The analysis is quite extensive. The authors performed study over multiple model architectures, ranging from smaller ones to larger ones, and evaluated various types of unsafe examples (bias, toxicity, harmfulness) and observe consistent findings.
- The proposed approach, ForgetFilter, is simple and effective. I appreciate author's evaluation of long-term safety where the proposed approach clearly outperforms alternative methods.

**Weaknesses:**

- I believe the authors should discuss related works that filter training data / perform data selection based on learning dynamics like frequency of forgetting.

Some papers:

[1] Maini et al. Characterizing Datapoints via Second-Split Forgetting, NeurIPS 2022

[2] Swayamdipta et al. Dataset Cartography: Mapping and Diagnosing Datasets with Training Dynamics. EMNLP 2020

- I wonder whether in practice "safe examples" are always readily available for a fine-tuning task. For example, if I download a random dataset D from the web without having a safe dataset of the same task as D, can I still apply ForgetFilter? Is this setup practical?

**Questions:**

- We see from Figure 2 that up to 80% of harmful examples will be forgotten in the end. But what is the proportion of harmful examples among all the forgotten examples?

I am asking the question above because I am a bit surprised to see applying ForgetFiltering improves downstream performance as well in Table 2. I thought ForgetFiltering may also remove training examples that are not harmful mistakenly. Can you explain side effects that is happening on downstream performance?

- See the "weakness" part for my question about applying ForgetFilter when no safe dataset is available.

---

> ### Author Response · Authors · 2023-11-16
> **Response to Reviewer pR1P**
>
> ## Response to Weakness 1:
> > I believe the authors should discuss related works that filter training data / perform data selection based on learning dynamics like frequency of forgetting.
>
> Thanks for pointing out two related works! We have added related works on data selection based on learning dynamics to our script.
>
>
> ## Response to Weakness 2:
> > I wonder whether in practice "safe examples" are always readily available for a fine-tuning task. For example, if I download a random dataset D from the web without having a safe dataset of the same task as D, can I still apply ForgetFilter? Is this setup practical?
>
> Our ForgetFilter works when unsafe data in D is not known. In that case, the safe dataset should be constructed to contain diverse kinds of safe examples, e.g., like the dataset that people use for aligning LLMs during pre-training.  Then the unsafe example whose categories are included by the safe dataset can be filtered. Such a safe dataset is commonly available in LLM companies and is ready  for users’ requests for their finetuning tasks.
>
> Additionally, the safe dataset does not need to be from the same or similar distribution of unsafe data in D. The forgetting is mainly based on the semantics of data (e.g., toxic or nontoxic).  For example, the unknown toxic data we experiment with are randomly sampled from web data and can be very different from untoxic data in our safe dataset.  However, toxic data can still be filtered through our proposed ForgetFilter.
>
> Overall, we think our setup is practical and our ForgetFilter is applicable to real-life scenarios.
>
> ## Response to Question 1:
> > We see from Figure 2 that up to 80% of harmful examples will be forgotten in the end. But what is the proportion of harmful examples among all the forgotten examples? … applying ForgetFiltering improves downstream performance … explain side effects that is happening on downstream performance?
>
> Thanks for your question. Among all the filtered examples, unsafe examples account for around 81.3% in all three cases (i.e., bias, toxicity and harmfulness).  Indeed, some training data of downstream tasks will be filtered out by mistake, but a lot more unsafe data that are irrelevant to the task are filtered. Then during training, the model will focus on learning downstream task data instead of unsafe examples.  We posit that this accounts for the marginal performance enhancement observed compared to directly fine-tuning on noisy downstream data.

---

### Official Review · Reviewer_qeKd · 2023-11-03

**Soundness:** 1 poor
**Presentation:** 2 fair
**Contribution:** 1 poor
**Rating:** 3
**Confidence:** 5

**Summary:**

The authors first analyze if the language models can forget unsafe content on fine-tuning. For this the authors first fine-tune a pre-trained language model on a noisy dataset with both safe and unsafe content and then fine-tune this model on safe data and analyze if the model is able to forget the classification of the unsafe data points in the first fine-tuning. Authors name the second fine-tuning stage as safety review. The author utilize different datasets to study the effect of presence of bias, toxicity and harmfulness, and conclude that during safety review the model can easily forget the unsafe content. Motivated by this, the authors propose a forget filter algorithm, which analyzes which samples are forgotten first during the safety review. The ones which are forgotten first are unsafe samples and therefore can be removed from the noisy dataset to make it safe. In the considered setup, the authors demonstrate that such a method could be used to filter out the data and make the models safer.

**Strengths:**

* The authors have attempted to make models safer and better aligned. This is an important question in the safety community.

* The experiments attempted by the authors are interesting

**Weaknesses:**

Major Weaknesses:

* The authors have only presented the case where there is presence of limited amount of safe and unsafe data. In real worlds the models are trained on internet scale corpus where the data is present from different domains and distributions. It is likely that on safety review the model would forget the out of distribution samples from the safety review finetuning dataset rather than explicitly forgetting the unsafe datapoints. Since the authors have limited their dataset to limited amount of safe and unsafe data points, it might give an illusion that the model can actually unlearn the unsafe datapoints. Therefore a more rigorous evaluation where the authors can also consider different sources and domains of safe and unsafe data, should be done.The scalability of the proposed method to filter the features from the pre-training dataset is not clear.

Minor weaknesses:

* The authors should try to perform some attacks like jailbreaking on the models fine tuned on the filtered dataset. If the unsafe content can be filtered from the dataset, then the model should not be jailbroken easily.

* A recent work [1] also uses the concept of second fine-tuning in order to filter out noisy samples. The idea of the proposed approach seems to be very similar to this. It would be nice, it the authors could look into this work [1].

[1] Maini, Pratyush et al. “Characterizing Datapoints via Second-Split Forgetting.” ArXiv abs/2210.15031

Update after rebuttal: I appreciate the authors for replying to my concerns. Unfortunately, authors have not addressed my concerns sufficiently. They have not provided additional evidence on jailbreaking attacks. Also I don't think that restricting the evaluation setup to fixed number of safe and unsafe datapoints can help in guaranteeing that the model can become safer using the proposed approach. Given these outstanding concerns I would like to maintain my score.

**Questions:**

I would request the authors to kindly address the comments in the weakness section.

---

> ### Author Response · Authors · 2023-11-16
> **Response to Reviewer qeKd (1/3)**
>
> Thanks for your insightful comments. We think that some concerns are caused by misunderstanding, which we will explain in detail below. We hope that our response can clarify the misunderstandings and you can consider our work more favorably.
> ## Response to Major weakness 1:
> 1. > The authors have only presented the case where there is presence of limited amount of safe and unsafe data ... filter the features from the pre-training dataset is not clear.
>
> First, we would like to point out that our work focuses on **customized finetuning**, instead of **pre-training**, to mitigate the safety issue of already pre-trained LLMs with RLHF and released to the public.
>
> During customized finetuning by normal users, as pointed out by reviewer LEME (``Considering that fine-tuning datasets are typically small’’),  the dataset tends to be smaller and more specific than internet-scale corpus for pre-training since users may focus on some specific downstream applications.  This is especially true when users use Web APIs to finetune LLMs by uploading their training data.
>
> The safety of LLMs in customized finetuning has emerged as a significant concern [1].  Users have the option to upload their data to the platform through APIs for fine-tuning, or they may choose to finetune open-sourced models with their collected data.  It's imperative to recognize that in both scenarios, the data used for fine-tuning has the potential to contain unsafe examples, whether intentional or unintentional, which can easily jailbreak the safety precaution of pre-trained LLMs (please see our response to minor weakness 1).
>
> All in all, we consider that our setting is both realistic and well-suited for investigating the safety issue of customized finetuning released LLMs.  While we acknowledge the importance of data quality in pre-training, it's essential to clarify that our work is not intended to address this particular issue.
>
>
>
> 2. > Therefore a more rigorous evaluation where the authors can also consider different sources and domains of safe and unsafe data, should be done.
>
> Thanks for your advice.  We point out that our experiments already consider the three most common aspects of unsafety into considerations, i.e., bias, toxicity, and harmfulness. Those data are very diverse. For example, the toxic and untoxic data are sampled from a subset (3338519 entries) of Pile [2] collected from web data of varied sources.
>
>
> 3. > It is likely that on safety review the model would forget the out of distribution samples from the safety review rather than explicitly forgetting the unsafe datapoints ... can actually unlearn the unsafe datapoints.
>
>
> Thanks for your question whether the unsafe data is explicitly forgotten through safety review.  We appreciate your mentioning "explicitly forgetting". To clarify, we interpret your statement as suggesting that a data point is essentially "explicitly forgotten" when the model exhibits behavior during training as if it has never encountered that specific data. Actually, in our interleaved training setup in Section 4.3, as one of our contributions, we reveal that unsafe examples may not be explicitly forgotten through supervised finetuning.  The LLM is continuously trained with noisy downstream data with safety review integrated into each session of downstream finetuning.  Our findings indicate that LLMs can rapidly reacquire previously "forgotten" malicious knowledge, seemingly transitioning to an unsafe mode. However, the initial exposure of LLMs to unsafe examples results in a slower learning process. This observation suggests that the unsafe examples are not explicitly forgotten during supervised finetuning on safe data.
>
> Additionally, we would like to clarify that safety review is not our highlighted contribution, but rather a method based on commonly used supervised finetuning so as to realign a pre-trained LLM that has been finetuned on malicious examples. Our proposed ForgetFilter performs much better than safety review in our interleaved training setup in Section 4.3, which implies the ability of ForgetFilter to ensure the long-term safety of LLMs.

---

> ### Author Response · Authors · 2023-11-16
> **Response to Reviewer qeKd (2/3)**
>
> ## Response to Minor weakness 1:
> > The authors should try to perform some attacks like jailbreaking … then the model should not be jailbroken easily.
>
> Thanks for your suggestion. We will explore adversarial prompts to bypass the safety alignment of LLMs in our future work.  This work mainly focuses on the issue of jailbreaking LLMs through finetuning with unsafe data. Our motivating use case is **released** LLMs can be trained on noisy data containing adversarial samples during users' customized finetuning.  In Section 3.2 and Figure 1, we finetune the aligned model with noisy data containing unsafe examples.  We find this can easily jailbreak the aligned model and elicit malicious generation.  Similar to your suggestion, in the "interleaved training" in Section 4.3, we show our filtering method is an important and effective way to ensure the model’s long-term safety in scenarios where unsafe and malicious data are repetitively and periodically presented to LLMs during downstream finetuning.
>
>
> ## Response to Minor weakness 2:
> > A recent work [3] also uses the concept of second fine-tuning in order to filter out noisy samples.
>
>
> Thanks for your pointing out a related work. We have added it to our Related Works section.  Despite the similar high-level concept and the phenomenon-level convergence of our results that both show the discrepancies in forgetting, our works are fundamentally different from this work.
>
> This work mainly focuses on forgetting patterns wrt. labels, i.e., data with correct labels and mislabeled ones.  The mislabeled ones are produced by randomly flipping labels. In contrast, our study is focused on forgetting wrt. the high-level semantics of data, i.e., the notion of safety. The concept of a “label” is not well defined in our case, since our data points are language sequences.   Unsafe examples, as opposed to safe ones, don't typically exhibit random inconsistent patterns like noisy mislabeled data.  Instead, they often contain systematic information, e.g., bias, or toxicity.
>
> Additionally, we include data that are more irrelevant to the notion of safety and investigate their forgetting patterns during safety finetuning as well. No past work has studied the forgetting patterns of data of different high-level semantics from diverse sources.

---

> ### Author Response · Authors · 2023-11-16
> **Response to Reviewer qeKd (3/3)**
>
> ## References:
> [1] Henderson, Peter, et al. "Self-destructing models: Increasing the costs of harmful dual uses of foundation models." Proceedings of the 2023 AAAI/ACM Conference on AI, Ethics, and Society. 2023.
> [2] Gao, Leo, et al. "The pile: An 800gb dataset of diverse text for language modeling." arXiv preprint arXiv:2101.00027 (2020).
> [3] Maini, Pratyush, et al. "Characterizing datapoints via second-split forgetting." Advances in Neural Information Processing Systems 35 (2022): 30044-30057.

---

### Author Response · Authors · 2023-11-16
**General response**

We appreciate all the reviewers’ time and efforts for providing reviews. We have updated our script with modifications highlighted in blue. We would like to clarify our motivations and contributions.

Our study aims to improve the safety issue of LLMs that are released to the public for downstream finetuning, not during pretraining.  A use-case is the public can upload their own data for finetuning customized LLMs through APIs, while data may contain malicious examples. There is a lack of understanding of safety issues in downstream finetuning and the effectiveness of safety precautions of released LLMs.

1. We first study the impact of unsafe examples in noisy downstream data and demonstrate that the safety precautions of released LLMs can be easily bypassed through supervised finetuning. We introduce a sequential safety review session to recover safety in finetuned LLMs.

2. We then investigate the forgetting patterns of LMs at different scales during the safety review. We
unveil the discrepancies in forgetting that for sufficiently large LMs, unsafe examples will be
forgotten more significantly than other examples in noisy downstream data when finetuned
with safe examples.  Such discrepancies in forgetting are found to be an **emergent** phenomenon for LLMs-only large-scale LLMs exhibit such behavior.

3. We propose ForgetFilter as an effective method to filter unsafe examples in noisy downstream data. Compared to safety review after downstream finetuning where the learned important downstream information can be forgotten, ForgetFilter will not compromise downstream task performance, while keeping the finetuned LLMs safe.

4. We further investigate “interleaved training” where downstream finetuning and safety review
are interleaved continuously. We demonstrate that LLMs can quickly reacquire previously
“forgotten” malicious knowledge and even produce more unsafe generations in the long run despite the safety review, highlighting the challenges for long-term safety assurance. However, our proposed ForgetFilter method can reduce the malicious generation in all sessions of downstream finetuning, which showcases the effectiveness of our method in improving the long-term safety of released LLMs.

Additionally, compared with the past works also leveraging forgetting patterns for filtering, our work is fundamentally different. Past works are based on forgetting in data with correct labels and mislabeled ones.  We, however,  unveil the forgetting patterns wrt. the high-level semantics of data, i.e., the notion of safety.

---

### Meta-Review · Area_Chair_Wp4G · 2023-12-03

**Metareview:**

The paper present ForgetFilt, an approach to identiy harmful training examples in order to reduce harm in learned models. While this is very interesting, the reviews also present in my opinion salient arguments about the suitability of this paper for ICLR in its current form: missing related work and an unclear usefulness of the setting considered. I fully agree. The concerns raised in the reviews need to be clarified before publication.

**Justification For Why Not Higher Score:**

The reviews raise a number of concerns, such as missing related work and an unclear usefulness of the setting considered.

**Justification For Why Not Lower Score:**

N/A

---

### Decision · Program_Chairs · 2024-01-16

Reject